# The Influence of Personality Type D and Coping Strategies on Cognitive Functioning in Students

**DOI:** 10.3390/bs14050382

**Published:** 2024-05-01

**Authors:** Alexey N. Sumin, Ingrid Yu. Prokashko, Anna V. Shcheglova

**Affiliations:** 1Federal State Budgetary Institution “Research Institute for Complex Issues of Cardiovascular Disease”, Blvd. Named Academician L.S. Barbarasha, 6, Kemerovo 650002, Russia; nura.karpovitch@yandex.ru; 2Federal State-Funded Educational Institution of Higher Education, Kemerovo State Medical University, Ministry of Health of the Russian Federation, Voroshilova Str., 22a, Kemerovo 650029, Russia; proing59@mail.ru

**Keywords:** medical students, type D personality, cognitive function, coping strategies

## Abstract

Introduction: Academic and emotional challenges faced by medical students can affect their psychological well-being and health. Personal characteristics may also predispose one to the manifestation of distress reactions. Individuals with type D personality have an increased tendency to develop depressive reactions and somatic diseases, including the presence of cognitive dysfunction. In students, the presence of cognitive dysfunction may additionally adversely affect academic and psycho-emotional problems. The purpose of this study was to examine the influence of type D personality and coping strategies on cognitive functioning in medical students. Methods: A cross-sectional study included 258 medical students (age 19 ± 1.2 years, 79 men). All participants completed psychological questionnaires (DS-14 to identify type D personality, and The Coping Strategy Indication, CSI—to determine coping strategies), as well as extensive neuropsychological testing of cognitive functions. Results: Among the medical students examined, the frequency of identification of type D personality was 44%. In persons with personality type D, according to psychometric testing, a decrease in the level of functional mobility of nervous processes (FMNP) was noted, which was manifested in an increase in the test completion time (*p* < 0.001) and an increase in the number of errors (*p* < 0.001) during the FMNP test, and an increase in the test completion time in the attention concentration test. In addition, in type D participants, an increase in the test execution time during the attention test was noted (*p* = 0.007). Personality type D was an independent risk factor for cognitive decline in students in multiple linear regression analysis, when type D was analyzed as a dichotomous construct. Conclusions: Assessing personal characteristics and identifying personality type D is advisable for medical students, to develop subsequent programs to increase their resistance to academic challenges, improve cognitive function, and also to prepare for future stress loads during professional activities in the field of healthcare.

## 1. Introduction

Long-term exposure to stress can result in students experiencing numerous stress symptoms, such as physical symptoms, anxiety disorder, depressive symptoms, and cognitive symptoms, which can also affect academic performance [1]. Mental health problems, such as depression and anxiety, are widespread among students, due to high stress loads during the acquisition of new knowledge [2,3]. The onset of stress begins very early in medical training and continues to be high throughout medical school and residency [4,5]. According to one study, 27.2% of medical students suffered from depression, and 11.1% had suicidal ideation [2]. Despite lower baseline rates of suicidality, suicide rates are higher among medical students just months after entering medical school compared to peers of the same age [2]. We can also mention relatively new problems among students—smartphone addiction and immersion in social networks [6,7]. Ultimately, such disorders not only negatively affect students’ academic performance, but also contribute to the development of diseases. Therefore, it is important to continue to study personality factors that predispose to manifestations of psychological distress.

Recently, the concept of personality type D has been proposed, which consists of a combination of negative affectivity and its suppression in social interactions [8]. Individuals with this personality type were found to be more likely to experience depressive symptoms [9], and on the other hand, the negative influence of personality type D on reactivity under stressors was demonstrated [10,11,12,13]. Individuals with type D exhibit atypical physiological reactions to acute psychological stress, especially to social-evaluative stressors [10]. Persons with type D experience an increased sense of distress and threat, and foresee great difficulties in forming a speech response in social situations. This leads to increased physiological arousal, meaning that people with type D exhibit atypical cardiovascular profiles when faced with stress [14]. Therefore, it is proposed to consider personality type D as a factor predisposing to the development of psychological distress.

It is not surprising that the study of type D personality in students revealed an association with a tendency to adverse consequences both in terms of mental health [2,3,15,16] and somatic diseases [16,17,18]. For example, in a study by Loas et al., the negative affectivity component of type D personality and anhedonia were shown to be independent predictors of suicidal ideation [15]. In addition, personality type D was one of the predictors of excessive involvement in social networks among students [6]. In addition, anhedonia and negative affectivity may represent two components of student dissatisfaction with the learning process, and ultimately with their chosen profession [15]. Personality type D significantly predicted emotional intelligence, resilience, and caring in female nursing students, which is an important component of nursing [19]. A significant connection was also found between personality type D and burnout syndrome among medical students [20].

The link between stress and coping in university students has previously been shown. Coping strategy is a key variable in the process of reducing, minimizing, or tolerating stress and preventing negative academic outcomes [21]. Coping strategies were characterized by avoidance strategies, such as cognitive avoidance, emotional release, and the search for alternative rewards. Students who experienced less stress and engaged in less cognitive avoidance and more problem solving had greater academic success; and students under more stress performed worse [21]. Our previous study demonstrated a link between type D personality and maladaptive coping strategies (escape-avoidance strategy) [22]. When performing a computer task with avatars, type D individuals had higher rates of avoidance in the avatar task than non-type D individuals [23]. Moreover, people with type D personality tend to use more passive and maladaptive avoidance strategies, such as resignation and withdrawal. This is associated with higher levels of perceived stress and increased levels of burnout symptoms [24]. Additionally, in athletes, type D was associated with increased levels of perceived stress, avoidance coping, and reduced performance (reduced distance; more errors) [25].

University students experience stress, and how they cope with this stress affects their academic performance [21]. Since cognitive processes are most actively involved during academic studies, an important task is to study the relationship between personality traits and cognitive function in students. In addition, academic burnout in adolescents was mitigated by cognitive reappraisal, highlighting the importance of a comprehensive approach encompassing cognitive, emotional, and physiological symptoms, to understand and address academic burnout among adolescents [26].

Among the mechanisms of stress, the following chain of reactions can be noted: the brain transforms the cognitive process of emotional stimuli into hemodynamic, neuroendocrine, and immune changes. Additionally, psychological stress is the body’s negative cognitive and emotional response to environmental demands that exceed the individual’s ability to cope [27]. It has previously been shown that there is a clear differentiation of the influence of type D personality on cognitive function between patients with cardiovascular diseases and healthy individuals. Cardiac patients with type D characteristics showed a marked reduction in specific cognitive planning function compared to non-type D patients, whereas this differentiation was not observed in a population group without cardiac disease [28]. At the same time, athletes with type D personality had a decrease in concentration [29] and difficulty completing new tasks [25]. Accordingly, the relationship between personality type D and the cognitive functioning of individuals requires in-depth study. Since both of these factors are components of the mechanism of stress influence on the cardiovascular system, one of the current directions is the examination of healthy individuals, including those with increased demands on the cognitive process during their studies.

We hypothesized that students with type D personality may have a decrease in cognitive functioning as measured by cognitive tests. The second hypothesis was that maladaptive coping strategies would also influence performance on cognitive tests. Accordingly, the purpose of the present study was to examine the influence of type D personality and coping strategies on the cognitive functioning of medical students.

## 2. Materials and Methods

### 2.1. Participants

This study involved 302 students of Kemerovo Medical University, males (n = 93) and females (n = 209) whose level of physical health according to G.L. Apanasenko [30] was no less than 4 points. The mean age of the participants was 18.7 years (standard deviation, SD = 1.8; range of 17–23), 25 students studied in the 1st year, and 277 in the second year. All studies were carried out in a laboratory in the morning (from 8.00 to 12.00), with stable positive health and performance, as well as a month or more after the completion of an exacerbation of a chronic, or the treatment of an acute, disease and at least 2 h after a light breakfast or on an empty stomach, and one hour after smoking.

### 2.2. Procedure

The study was carried out once, in the autumn. In accordance with the Declaration of Helsinki, written informed consent was obtained from each participant. All participants completed psychological questionnaires (DS-14 to identify type D personality, and The Coping Strategy Indication, CSI—to determine coping strategies), as well as extensive neuropsychological testing of cognitive functions. The processing of data complied with current regulations that guarantee the anonymity and security of information in general. When introducing students to the work of the department’s scientific laboratory, they were offered voluntary participation in the study. Because the subjects were university students, they were recruited by teachers with whom they had no academic interaction; participation in the study was voluntary and without any incentives.

Academic performance was assessed by calculating grade point average. Calculation of the average score was performed by adding the exam scores for each subject in the session and dividing by the total number of exams.

### 2.3. Measures

To identify personality type D, we used the DS-14 [8] questionnaire in the Russian version [31]. In this study, Cronbach’s alpha for NA was 0.78, and for SI, it was 0.74; which confirmed the adequacy of the intrinsic structure of the Russian version of DS14. The questionnaire contains 14 questions, which are divided into 2 scales: negative affectivity (NA) of emotion and the degree of social inhibition (SI). If the subject scores 10 or more points in each scale, then personality type D is determined. Answers to questions range from “false” (0 points) to “completely true” (4 points), including intermediate answers “rather false” (1 point), “hard to say” (2 points), “probably true” (3 points).

To assess coping strategies, the questionnaire “The Coping Strategy Indication” (CSI) [32], adapted into Russian by Sirota et al., was used [33], and Cronbach’s alpha mean for all scales was 0.78 (range from 0.81 to 0.75). The indicator of stress coping strategies allows us to distinguish three groups of fundamental coping strategies: a problem-solving strategy, a strategy of seeking social support, and an avoidance strategy. The questionnaire contains 33 judgments, to which the respondent gives the following answer options: completely agree, agree, and disagree. Answers are scored on a 3-point system. The scales have different levels of use of the individual’s dominant coping strategies: very low, low, medium, and high. We previously described in detail the application and interpretation of this questionnaire [34].

Extended neuropsychological testing of cognitive functions was performed using the psychophysiological complex “STATUS PF” (certificate No. 2001610233 of the Russian Patent and Trademark Agency) [35], which included assessment of cognitive domains: attention, memory, and neurodynamics (Table 1). To encourage a positive attitude of the subjects towards the psychophysiological examination, the meaning and significance of the research was first explained to them. Based on the test results, individual changes in neuropsychological indicators were calculated for each patient.

The study of the complex visual-motor reaction (SVMR), the choice reaction, was carried out under conditions of selective response to various stimuli that differed in shape, color, size, and other characteristics. The principle of this method is to measure the speed of reaction to a light signal and study the stability of nervous processes. Colors appeared on the screen in a chaotic order (there were 30 signals in total); the subject’s task was to respond to red (with his right hand) and green (with his left hand). An important component of the instructions was to respond as quickly as possible. The subject was not supposed to react to the yellow color. To avoid habituation—reducing the reaction to repeated stimuli, light signals were given chaotically, but they were regular, each next signal was expected. The interval between signals ranged from 0.5 to 2.5 s.

When studying the complex visual-motor reaction (CVMR), the minimum and average exposure (ms) of the response time to the stimulus and the number of errors were recorded. In healthy individuals, these values were in the range of 400–425 ms [36]. A short reaction time reflects a high quality of neuronal activity in the associative areas of the prefrontal cortex, which are responsible not only for the analysis of sensory signals, but also for the organization of motor reactions in response to them. The physical limit or “irreducible minimum” of stimulus response time is about 100 ms. In addition, performing CVMR is associated with certain volitional efforts, which reflects the level of nonspecific and specific activation of the central nervous system. Successful performance of a reflexometric task requires an optimal level of activation [37].

The assessment of the functional mobility of nervous processes (FMNP) was carried out in the “feedback” mode. To process information, 120 different color stimuli were offered. The duration of exposure to the testing signal changed automatically depending on the nature of the patient’s responses. The range of fluctuations in signal exposure was in the range of 200–900 ms. The sequence of signal presentation was random, while maintaining equal representation of each species. The test execution time (sec), the minimum signal exposure value (ms), the time to reach the minimum exposure (sec), the average exposure (ms), the number of errors made, and the number of missed positive signals were recorded. For students with a high level of functional mobility of nervous processes, the time to complete the test was ≤60 s, with an average level of FMNP—61–70 s, and with low FMNP ≥ 71 s.

To study voluntary attention, a technique was used to determine concentration of attention. The subject was presented with 10 lines of numbers, and had to carefully but quickly looking through the lines and then highlight adjacent numbers in one line (one line has 33 numbers), which add up to 10. The time to complete the test for concentration of attention (seconds) and the number of correct answers as a percentage were determined. High concentration of attention had 10% of errors, good concentration had 20%, satisfactory data had 30%, and poor concentration had 40% of errors.

To study visual memory, the “10 numbers memorizing test” technique was used. The principle of this method is that 10 different numbers appear sequentially on the screen, which the subject must remember and reproduce in any sequence. The criteria for assessing the level of visual memory are as follows: first level (very low) 0–2 reproduced numbers, second level (low) 3–4 reproduced numbers, third level (medium) 5–6 reproduced numbers, fourth level (high) 7–8 reproduced numbers, and the fifth level (very high) 9–10 numbers reproduced.

To study verbal–logical memory, the “10 words memorizing test” technique was used. The subject had to remember and reproduce in any sequence 10 different words appearing on the monitor screen. The number of words reproduced was recorded and the level of verbal-logical memory was determined: 0–2—very low; 3–4—low; 5–6—average; 7–8—high; 9–10—very high. The words were displayed on the screen at intervals of 2–3 s in a chaotic order.

### 2.4. Statistical Analysis

For statistical processing, the programs “STATISTICA 8.0” (Dell Software, Inc., Round Rock, TX, USA) and SPSS 17.0 (IBM, Armonk, New York, NY, USA) were used. The distribution of quantitative variables was checked for normality using the Kolmogorov–Smirnov test. With a normal distribution, quantitative indicators are presented as means ± standard deviation; if the distribution is different from the normal one—in the form of the median and quartiles (25th and 75th percentiles). A Student’s *t*-test, Mann–Whitney test, and χ2 (chi-square) test were used to compare two groups (with type D and without type D). To assess cognitive functions associated with the presence of type D personality, multiple logistic regression analysis was performed (Forward Stepwise LR method). Demographic factors, indicators of cognitive function, and data from a questionnaire assessing coping strategies were included in the models as independent variables. To identify a possible association of the cognitive tests indicators (“concentration of attention” test and “functional mobility of nervous processes” test) with components of personality type D (type D, ZscoreNA, ZscoreSI, and zNA x zSI), we additionally performed linear regression analysis. The critical significance level (*p*) was set at 0.05.

## 3. Results

A total of 258 people completed all psychological and cognitive tests that were suitable for further analysis. Students were divided into two groups: without type D (n = 144) and with type D (n = 114). The general characteristics of the studied groups are presented in Table 2. The groups were comparable in age (*p* = 0.371). There were significantly fewer young men in the group with type D than in the group with non-type D (*p* < 0.001). Students with type D personality had significantly higher average scores on the NA and SI scales than students without type D (*p* < 0.001 in both cases). Students with type D in 83.02% of cases gave a positive subjective assessment of the influence of personality type D on academic performance, and in the group without D in 67.3% (*p* = 0.166). However, the groups were comparable in terms of academic performance (*p* = 0.166).

Figure 1 presents a comparative analysis of the dominant coping strategies according to the CSI questionnaire, depending on the presence or absence of personality type D in medical students. Students with type D personality were less likely to use problem-solving strategies (56.8% and 76.3%, *p* = 0.02) compared to students without type D. There were no significant differences in other strategies.

The data from neuropsychological testing in medical students are presented in Table 3. According to the results of the CVMR, no differences were identified between the groups. Exposure indicators were within normal limits.

The groups differed significantly in the nature of the students’ response to the presented sensorimotor loads during the FMNP test. All students had a high level of functional mobility of nervous processes (the time to complete the test was >61 s). Meanwhile, students with type D took significantly longer to complete the test (*p* < 0.001), with a longer time to reach the minimum exposure (*p* = 0.046) and average exposure (*p* = 0.042) than students without type D. At the same time, the number of errors made in the group with type D was greater (27.5 versus 24.0, respectively; *p* < 0.001), and the number of missed positive signals was less (7.0 versus 9.0, respectively; *p* = 0.03).

Analysis of the results of the “Concentration of Attention” test demonstrated good concentration in both groups (number of errors > 20%). However, students with type D spent significantly more time passing the test (*p* = 0.0071), but at the same time made fewer errors in the test than students without type D (12.1%, versus 17.5%; *p* < 0.001).

Tests for “10 numbers memorizing” and “10 words memorizing” in the study groups did not have significant differences.

In two binary logistic regression models (direct LR method), we studied the association of indicators of psychometric tests with the presence of personality type D. In the first of them, the association of indicators of the tests “Complex visual-motor reaction” and “Memory” with type D was assessed. An independent association was found for the following factors (χ2 (2) = 17.988; *p* < 0.001): minimal exposure with CVMR (B = −0.011; *p* = 0.015), and female gender (B = 1.395; *p* = 0.004). This model explained only 22.2% (Nagelkerke R^2^) of the variance in personality type D and correctly classified 70.3% of cases (Table 4). In the second model, associations of data from the “functional mobility of nervous processes” and “Attention” tests with personality type D were studied. An independent association was identified for the following factors (χ2 (4) = 26.787; *p* < 0.001): test execution time (B = 0.094; *p* = 0.005) and number of mistakes with FMNP (B = 0.128; *p* = 0.004), number of correct answers with the attention test (B = 0.064; *p* = 0.028). This model explained only 31.0% (Nagelkerke R^2^) of the variance in personality type D and correctly classified 79.1% of cases (Table 5).

Additionally, we assessed the association of indicators of the DS-14 questionnaires (personality type D, values on the NA and SI scales, as well as the possible synergistic effect of these subscales) with the values of cognitive tests using multiple linear regression analysis. Significant associations with personality type D were only identified for test execution time (concentration of attention test) (Beta = 0.371, *p* = 0.015) (Table 6). The functional mobility of nervous processes test revealed the following associations: the average exposure with Zscore (SI) (Beta = 0.288, *p* = 0.039) and the number of missed positive signals with Zscore (NA) (Beta = −0.241, *p* = 0.047) (Table 7). In no cases was a synergistic effect of the NA and SI subscales on psychometric test scores identified.

## 4. Discussion

The present study examined the relationship between the presence of type D personality and cognitive test scores in healthy young adults. To our knowledge, this is the first study to examine this issue in medical students. According to psychometric testing, the level of functional mobility of nervous processes that suffered most in type D was manifested by an increase in test completion time and the number of errors during the test. In addition, in type D, an increase in test execution time during the attention test was noted. In addition, we found that a number of psychometric test scores were independently associated with type D personality as a dichotomous construct.

Negative affectivity, social inhibition, and ineffective coping strategies characteristic of type D personality may contribute to impairment of general cognitive function. However, it is not always possible to identify an association between type D personality traits and cognitive function in healthy individuals. Thus, in the Gutenberg Health Study [28], in persons without cardiovascular pathology, it was not possible to detect a decrease in cognitive function, which was determined by the ability to plan when performing the Tower of London test. In addition, the influence of personality type D on cognitive function has not previously been assessed in athletes, but some signs of its dysfunction have been noted (decreased concentration, difficulties in performing new tasks) [25,29]. In the present study, we used a more complex battery of tests to assess cognitive function, which may have been the reason for identifying cognitive dysfunction in individuals with type D personality. The dependence we identified requires confirmation in subsequent studies in healthy individuals. At the same time, for patients with cardiovascular diseases, an association of personality type D has already been shown with the presence of cognitive dysfunction [28,38,39]. The presence of cognitive dysfunction in medical students may contribute to worse academic performance (however, this could not be detected in the present study) or to greater stress load during the study process. Accordingly, type D personality is a risk factor for burnout among medical students given the high level of academic workload.

A study of coping strategies showed that students with type D were less likely to use problem-solving strategies compared to students without type D. However, the hypothesis about the association of inadequate coping strategies with the presence of cognitive dysfunction according to cognitive tests was not confirmed in the present study. This is somewhat at odds with previous research, which found that students who were more likely to use problem-solving strategies had greater academic success [21]. Apparently, further research on this topic should use not only an expanded battery of psychophysiological tests, but also a more detailed assessment of coping strategies (for example, using the Ways of Coping Questionnaire [22,40]).

In this article, we primarily considered personality type D as a dichotomous variable, as was originally proposed by the developers of the concept of this personality type [8]. Recently, it has been proposed to evaluate the “type D effect” either by assessing the continuous values of its subscales, as well as the synergistic effect of these subscales [41]. Therefore, we purposefully studied the association of all these indicators with the results of psychophysiological tests. For some tests, it was possible to identify an association with the presence of personality type D, for others—with its individual subscales, but in no case was a synergistic effect of these subscales revealed, which is quite consistent with the data of previous studies [42,43]. We believe that, in future studies, it is legitimate to use the classic version of assessing personality type D, as has been shown both by this study and by previous works in this direction [22,44].

We see the significance of this study in identifying from among medical students a group of people who are most susceptible to academic stress and burnout due, among other things, to the presence of psychological personality characteristics and a tendency to cognitive dysfunction. If it can be shown that, with type D personality, it is possible to correct not only the personal characteristics of students, but also their cognitive functioning, then this will be a way to reduce stress loads during their studies. It must be said that a number of studies have been able to show this. Thus, in middle-aged women in Korea, lifestyle interventions (adjusting diet, physical activity, stress management, and improving cognitive function) influenced both the manifestations of type D personality and cognitive function [45]. Interestingly, improvements in cognitive function and decreases in type D traits occurred simultaneously. This makes us think about the possible existence of an inverse relationship between cognitive functions and type D personality; however, this assumption requires confirmation in further research. There is an example of a program to combat distress in medical students [46], known as “Mindfulness-Based Attention Training” for medical students. Participants self-reported that the program contributed to improvements in academic achievement, concentration, interpersonal relationships, and psychological health [46]. It is possible that it is for students with type D personality that such a program would be most effective, but this requires further research in this area.

Finally, it should be remembered that type D personality is associated with fatigue and high levels of burnout at work among medical personnel [47,48], affecting the achievement of professional goals [19], especially those working in critical care and surgery [48]. Therefore, among medical students with type D personality, interventions aimed at correcting maladaptive coping strategies [49] and special cognitive training [50] can not only reduce the level of stress during their studies, but can also subsequently contribute to more successful professional activities.

### Study Limitations

There were several limitations to this study. First, it was carried out in a single center with a relatively small number of individuals examined. Larger multicenter studies are needed to test the effects of type D personality on cognitive test scores. Second, this was a cross-sectional study, so a causal relationship between type D personality and cognitive dysfunction cannot be inferred. Third, we assessed personality type D as a dichotomous variable, although more recent studies have proposed treating the construct of personality type D as a continuous variable rather than a dichotomous one. However, the dichotomous version of personality type assessment is a convenient, albeit somewhat simplified, construct that has proven its value in clinical settings, showing its prognostic value and impact on the quality of life of patients. In addition, our results on the association of type D with cognitive function concern only those tests that we studied. It cannot be excluded that this association will not be confirmed when assessing other cognitive functions.

It is important to acknowledge that a significant difference in participation between genders was observed. This is quite understandable, there are specific features of the gender composition of medical university students—in general, there are more female students than male students. In addition, the greater involvement of girls in scientific projects may be explained by their socio-psychological characteristics. Female medical university students are characterized by such traits as conscientiousness, responsibility, and commitment, which explains their educational motivation and greater participation in scientific projects compared to male students. Nevertheless, the significant difference in participation between genders, with a predominance of female students, is a potential limitation of the study.

## 5. Conclusions

Among the medical students examined, the frequency of identification of type D personality was 44%. In persons with personality type D, according to psychometric testing, a decrease in the level of functional mobility of nervous processes was noted, which was manifested in an increase in the test completion time and an increase in the number of errors during the FMNP test, and an increase in the test completion time in the attention concentration test. We also found that personality type D was an independent risk factor for cognitive decline in students in multiple linear regression analysis, when type D was analyzed as a dichotomous construct. Assessing personal characteristics and identifying personality type D is advisable for medical students, to develop subsequent programs to increase their resistance to academic challenges, improve cognitive function, and also to prepare for future stress loads during professional activities in the field of healthcare.

## Figures and Tables

**Figure 1 behavsci-14-00382-f001:**
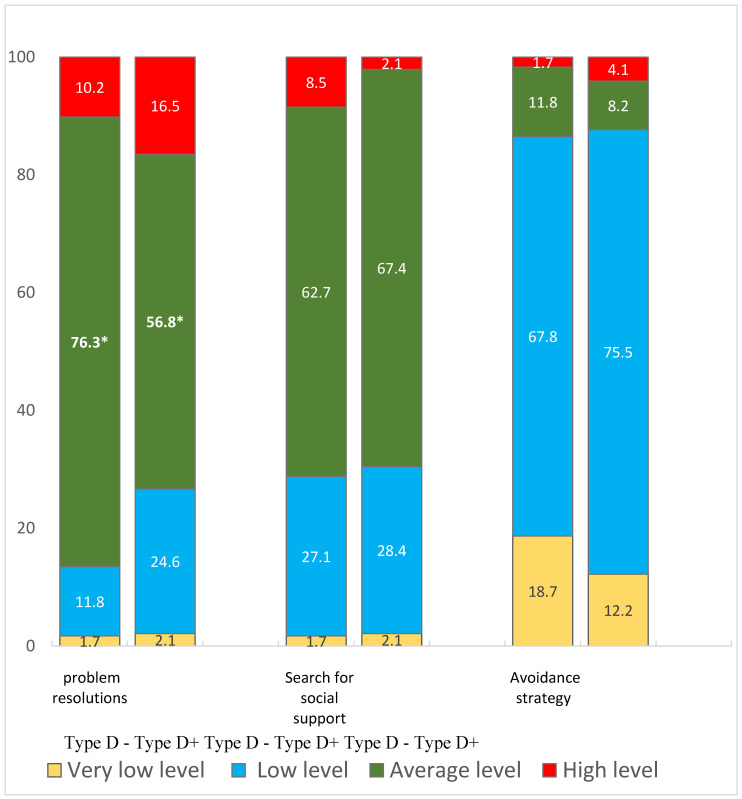
Comparative analysis of dominant coping strategies for medical students with presence/absence of personality type D (according to the CSI questionnaire). * *p* = 0.02 between group with/without type D.

**Table 1 behavsci-14-00382-t001:** Tests for studying cognitive functions in medical students.

Cognitive Tests	Methodology	Values
Complex visual motor response (CVMR)	Selective response to various stimuli that differ in shape, color, size, and other characteristics. The subject’s task is to react with his right and left hand when a signal appears (rectangles of different colors) under the conditions of choosing one of three presented signals (the number of signals in the test is 30)	An increase in the average latent period of the reaction and an increase in the number of errors during repeated studies indicate a deterioration in the functional state of the central nervous system
Level of functional mobility of nervous processes (FMNP)	The test was carried out in feedback mode. The exposure duration of the testing signal changed automatically after a correct answer; the exposure of the next signal was shortened by 20 ms, and after an incorrect answer it was lengthened by 20 ms (the number of signals in the test is 120)	An increase in the number of missed signals and the number of errors during repeated studies indicates a violation of the mobility of nervous processes
Concentration of attention	The test was carried out using a digital array presented to the test subject, from which it was necessary to select pairs of adjacent digits, totaling 10 (in 7 min)	A decrease in the number of correct answers (≥60%) indicates poor concentration
10 numbers memorizing test	It is necessary to remember as many of the 10 numbers as possible, presented one after another	A decrease in the number of remembered numbers indicates a deterioration in the processes of short-term symbolic memory
10 words memorizing test	It is necessary to remember as many of the 10 words as possible, presented one after another	A decrease in the number of words remembered indicates a deterioration in the processes of short-term verbal-logical memory

**Table 2 behavsci-14-00382-t002:** General characteristics of medical students with the presence/absence of personality type D.

Indicators	Non Type D (n = 144)	Type D (n = 114)	*p*
Age, years (Me [Q1–Q3])	19.0 (18.0; 19.0)	19.0 (18.0; 20.0)	0.371
Men (n, %)	56 (38.9)	23 (20.18)	0.001
Academic performance, score (Me [Q1–Q3])	4 (4.0; 5.0)	4 (4.0; 5.0)	0.496
**DS-14**			
Negative affectivity (NA), points (Me [Q1–Q3])	7.0 (5.0; 10.0)	14.0 (12.0; 18.0)	<0.001
Negative affectivity (NA), points (M ± SD)	8.07 ± 4.01	15.25 ± 4.07	<0.001
Social inhibition (SI), points (Me [Q1–Q3])	8.0 (7.0; 10.5)	14.0 (12.0; 17.0)	<0.001
Social inhibition (SI), points (M ± SD)	8.69 ± 3.93	14.65 ± 3.59	<0.001
**Amirkhan’s Coping Strategy Indicator (CSI)**			
Problem-solving strategy, points (M ± SD)	25.1 ± 3.69	24.6 ± 4.48	0.37
Problem-solving strategy, points (Me [Q1–Q3])	25.0 (22.0; 28.0)	24.0 (21.0; 27.0)	0.37
Social support search strategy, points (M ± SD)	21.83 ± 4.8	20.94 ± 3.73	0.41
Social support search strategy, points (Me [Q1–Q3])	22.0 (18.0; 25.0)	21.0 (18.0; 23.0)	0.41
Avoidance strategy, points (M ± SD)	18.8 ± 3.41	19.06 ± 3.54	0.92
Avoidance strategy, points (Me [Q1–Q3])	19.0 (17.0; 21.0)	19.0 (16.0; 21.0)	0.92

**Table 3 behavsci-14-00382-t003:** Indicators of cognitive functions according to psychometric testing in medical students with/without personality type D.

Indicators	Non Type D (n = 144)	Type D (n = 114)	*p*
**Neurodynamics**			
*Complex visual-motor reaction*			
Minimal exposure, ms (Me [Q1–Q3])	111.0 (105.0; 117.0)	112.0 (107.0; 116.0)	0.487
Average exposure, ms (Me [Q1–Q3])	469.0 (440.0; 540.0)	452.0 (418.0; 490.0)	0.126
Number of mistakes (Me [Q1–Q3])	1.0 (0; 2.0)	1.0 (0; 2.0)	0.509
*Level of functional mobility of nervous processes*			
Test execution time, sec (Me [Q1–Q3])	56.0 (54.0; 60.0)	60.5 (57.0; 65.5)	<0.001
Minimum exposure, ms (Me [Q1–Q3])	187.0 (156.0; 234.0)	172.0 (156.0; 211.0)	0.44
Time to reach minimum exposure, sec (Me [Q1–Q3])	34.5 (27.0; 44.0)	43.5 (28.5; 53.0)	0.046
Average exposure, ms (Me [Q1–Q3])	350.5 (331.0; 364.0)	357,5 (343.5; 388.0)	0.042
Number of mistakes (Me [Q1–Q3])	24.0 (22.0; 27.0)	27.5 (24.5; 31.0)	<0.001
Number of missed positive signals (Me [Q1–Q3])	9.0 (6.0; 12.0)	7.0 (2.0; 11.0)	0.03
**Attention test**			
Test execution time, sec (Me [Q1–Q3])	184.5 (168.0; 210.0)	209.5 (183.0; 232.0)	0.0071
Number of correct answers, % (Me [Q1–Q3])	82.5 (75.59; 89.3)	87.87 (83.54; 94.35)	0.0013
**Memory test**			
10 numbers memorizing test, points (Me [Q1–Q3])	6.0 (5.0; 7.0)	6.0 (5.0; 7.0)	1.0
10 numbers memorizing test, level (Me [Q1–Q3])	3.0 (3.0; 4.0)	3.0 (3.0; 4.0)	0.972
10 words memorizing test, points (Me [Q1–Q3])	6.0 (5.0; 7.0)	6.5 (6.0; 8.0)	0.055
10 words memorizing test, level (Me [Q1–Q3])	3.0 (3.0; 4.0)	3.5 (3.0; 4.0)	0.336

**Table 4 behavsci-14-00382-t004:** Association of indicators of cognitive tests (complex visual-motor reaction and memory), gender and age with the presence of personality type D (binary logistic regression analysis, forward likelihood ratio).

		B	S.E.	Wald	df	Sig.	Exp (B)
Step 1	Female	1.417	0.471	9.047	1	0.003	4.125
	Constant	−2.834	0.829	11.677	1	0.001	0.059
Step 2	Female	1.395	0.486	8.223	1	0.004	4.035
	CVMR, minimal exposure	−0.011	0.005	5.960	1	0.015	0.989
	Constant	0.828	1.666	0.247	1	0.619	2.289

**Table 5 behavsci-14-00382-t005:** Association of indicators of cognitive tests (level of functional mobility of nervous processes and attention) with the presence of personality type D (binary logistic regression analysis, forward likelihood ratio).

		B	S.E.	Wald	df	Sig.	Exp (B)
Step 1	FMNP, number of mistakes	0.141	0.046	9.311	1	0.002	1.151
	Constant	−3.926	1.227	10.241	1	0.001	0.020
Step 2	FMNP, test execution time	0.101	0.033	9.183	1	0.002	1.106
	FMNP, number of mistakes	0.144	0.043	11.244	1	0.001	1.155
	Constant	−9.969	2.489	16.040	1	0.000	0.000
Step 3	Attention, number of correct answers	0.064	0.029	4.806	1	0.028	1.066
	FMNP, Test execution time	0.094	0.034	7.753	1	0.005	1.099
	FMNP, number of mistakes	0.128	0.045	8.188	1	0.004	1.136
	Constant	−14.607	3.453	17.900	1	0.000	0.000

**Table 6 behavsci-14-00382-t006:** Association of indicators of the “concentration of attention” test with components of personality type D (multiple linear regression analysis).

Model	Unstandardized Coefficients		Standardized Coefficients		
	B	Std. Error	Beta	t	Sig.
Test execution time					
(Constant)	190.038	7.866		24.158	0.000
Type D	33.510	13.590	0.371	2.466	0.015
Zscore (NA)	−11.233	7.198	−0.185	−1.561	0.122
Zscore (SI)	−0.216	5.672	−0.005	−0.038	0.970
zNA x zSI	−5.445	5.485	−0.099	−0.993	0.323
Number of correct answers					
(Constant)	82.788	1.489		55.584	0.000
Type D	4.846	2.573	0.280	1.884	0.063
Zscore (NA)	−1.598	1.363	−0.137	−1.172	0.244
Zscore (SI)	1.181	1.074	0.151	1.099	0.274
zNA x zSI	−0.194	1.038	−0.018	−0.186	0.853

**Table 7 behavsci-14-00382-t007:** Association of indicators of the “functional mobility of nervous processes” test with components of personality type D (multiple linear regression analysis).

Model	Unstandardized Coefficients		Standardized Coefficients		
	B	Std. Error	Beta	t	Sig.
Time to reach minimum exposure					
(Constant)	204.904	7.433		27.566	0.000
Type D	−21.975	12.841	−0.266	−1.711	0.090
Zscore (NA)	6.477	6.802	0.117	0.952	0.343
Zscore (SI)	6.030	5.359	0.162	1.125	0.263
zNA x zSI	0.036	5.183	0.001	0.007	0.994
Average exposure					
(Constant)	361.083	5.289		68.275	0.000
Type D	−4.884	9.136	−0.080	−0.535	0.594
Zscore (NA)	6.885	4.839	0.167	1.423	0.158
Zscore (SI)	7.981	3.813	0.288	2.093	0.039
zNA x zSI	1.065	3.687	0.028	0.289	0.773
Number of missed positive signals					
(Constant)	8.448	1.019		8.294	0.000
Type D	−0.628	1.760	−0.054	−0.357	0.722
Zscore (NA)	−1.871	0.932	−0.241	−2.007	0.047
Zscore (SI)	0.064	0.734	0.012	0.088	0.930
zNA x zSI	−0.183	0.710	−0.026	−0.258	0.797

## Data Availability

Data regarding this manuscript are available in the Federal State Budgetary Scientific Institution “Research Institute for Complex Issues of Cardiovascular Disease”, Kemerovo, Russia.

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
