# Peer review of "The Influence of Personality Type D and Coping Strategies on Cognitive Functioning in Students"

_behavsci, 2024, doi:10.3390/bs14050382_

Round 1
Reviewer 1 Report
Comments and Suggestions for Authors
Dear authors,
I hope this message finds you well. I am writing to you as a reviewer of the manuscript entitled "The Influence of Personality Type D and Coping Strategies on Cognitive Functioning in Students," which I had the privilege of reviewing recently.
First of all, I want to express my sincere appreciation for the effort and dedication you have put into your research. It is evident that you have done meticulous and significant work in an area that deserves considerable attention.
However, after carefully reviewing the content of the article, I would like to make some suggestions that I believe could improve the clarity and relevance of your research:
The introductory section focuses largely on the relationship between type D personality and cardiovascular disease, whereas the aim of the study focuses on the influence of this personality and coping strategies on the cognitive functioning of medical students. I suggest that you reconsider the relevance of the information provided in the introduction to ensure that it is more aligned with the main objective of the study. It would be beneficial to reduce the emphasis on the relationship with cardiovascular disease and focus more directly on the relationship between Type D personality and cognitive functioning in the context of medical students; something that is also perhaps somewhat lacking...due to the somewhat scarce number of studies focused on medical students.
On the other hand, I noticed a discrepancy between the number of participants mentioned in the participants section and the abstract of the article. While in the participants section they indicate that 302 subjects participated, in the abstract they mention 258. I suggest that they clarify and correct this discrepancy to avoid confusion and ensure consistency in the presentation of the data. Furthermore, in the section on participants, it would be beneficial to include additional information on the characteristics of the subjects, such as the ages included and the distribution by grades (e.g., how many belonged to the first grade and which ones belonged to the second grade). In the procedure subsection, I suggest that you provide details on how the recruitment of participants was conducted and where the tests were performed. Finally, in the subsection referring to the instruments, it would be useful to state the reliability obtained for each instrument by means of some type of statistical method such as Cronbach's alpha.
Finally, it would be necessary to further relate the results obtained with the existing scientific evidence in the discussion section with the recommendations set out in the suggestions on the introduction.
This series of changes could make the article more solid and improve its academic quality.
Best regards.
Author Response
I hope this message finds you well. I am writing to you as a reviewer of the manuscript entitled "The Influence of Personality Type D and Coping Strategies on Cognitive Functioning in Students," which I had the privilege of reviewing recently.
First of all, I want to express my sincere appreciation for the effort and dedication you have put into your research. It is evident that you have done meticulous and significant work in an area that deserves considerable attention.
Answer:
We are grateful to the reviewer for his careful review of our manuscript and helpful comments that pointed us to ways to improve the text of our manuscript.
However, after carefully reviewing the content of the article, I would like to make some suggestions that I believe could improve the clarity and relevance of your research:
The introductory section focuses largely on the relationship between type D personality and cardiovascular disease, whereas the aim of the study focuses on the influence of this personality and coping strategies on the cognitive functioning of medical students. I suggest that you reconsider the relevance of the information provided in the introduction to ensure that it is more aligned with the main objective of the study. It would be beneficial to reduce the emphasis on the relationship with cardiovascular disease and focus more directly on the relationship between Type D personality and cognitive functioning in the context of medical students; something that is also perhaps somewhat lacking...due to the somewhat scarce number of studies focused on medical students.
Answer:
Thank you for the suggestion, we have significantly adjusted the Introduction section and focused on the problems of students.
On the other hand, I noticed a discrepancy between the number of participants mentioned in the participants section and the abstract of the article. While in the participants section they indicate that 302 subjects participated, in the abstract they mention 258. I suggest that they clarify and correct this discrepancy to avoid confusion and ensure consistency in the presentation of the data. Furthermore, in the section on participants, it would be beneficial to include additional information on the characteristics of the subjects, such as the ages included and the distribution by grades (e.g., how many belonged to the first grade and which ones belonged to the second grade). In the procedure subsection, I suggest that you provide details on how the recruitment of participants was conducted and where the tests were performed. Finally, in the subsection referring to the instruments, it would be useful to state the reliability obtained for each instrument by means of some type of statistical method such as Cronbach's alpha.
Answer:
Initially, 302 students were included in the study (this is indicated in the Methods section), but 258 people completed all tests, so 258 students were included in further analysis (this is noted in the Results section).
We added to the text of the manuscript information about the age of the participants, their distribution among courses, and also included information about the procedure for recruiting study participants.
We have added information about Cronbach’s alpha for the Russian versions of DS14 and questionnaire “The Coping Strategy Indication” to the text of the manuscript.
Finally, it would be necessary to further relate the results obtained with the existing scientific evidence in the discussion section with the recommendations set out in the suggestions on the introduction.
Answer:
Thank you for your comment. We have made a significant correction to the Discussion section
This series of changes could make the article more solid and improve its academic quality.
Reviewer 2 Report
Comments and Suggestions for Authors
This study aimed to examine the influence of type D personality and coping strategies on cognitive functioning in medical students. The current study is on a topic of relevance and general interest to the readers of the journal. However, several issues need to be addressed:
1. The background section provides adequate information. However, there is an absence of the study's significance or research gap. To address this issue, it is suggested to reduce the length of the background and allocate more space to the existing research gap. This approach will help to emphasize the importance and relevance of the study, thereby making it more compelling and informative.
2. The manuscript title mentions Coping Strategies as an independent variable to explore its impact on cognitive functioning. However, the introduction, results, and discussion presented in the manuscript solely pertain to type D, leaving no indication of any findings related to Coping Strategies.
3. The introductory section appears to be inadequately written, and the subject matter appears to lack coherence and logic. The lack of coherence may have a negative impact on the overall quality of the work, and could potentially detract from the intended message. It is recommended that revisions be made to the introductory section to enhance its clarity, cohesiveness, and logical flow. By doing so, the overall quality and impact of the work may be improved. The initial statement appears to be merely informative and lacks a clear backdrop for a specific hypothesis.
4. The authors seem to have put an excessive focus on cardiovascular disease in their research, which may detract from their primary subject of medical students. It would be beneficial for them to include relevant studies that focus on the unique condition of healthy individuals, particularly students, to provide a thorough background for their research. Without a clear research hypothesis and questions, readers may struggle to understand the purpose of the study. The manuscript could benefit from a well-defined hypothesis and participants that align with the focus on medical students, rather than CVD patients.
5. In the methods section, it is recommended to include information regarding the validity and reliability of the measurement instruments used.
6. Please elaborate on the concept of individual meta-analysis as mentioned in page 2, line 47.
7. It is important to acknowledge that a significant difference in participation between genders has been observed. This finding must be justified and properly acknowledged as a potential limitation of the study.
8. It is imperative to acknowledge the origin of the procedure being discussed. Can we expect a replication study similar to the references provided in the following paragraph? Alternatively, are the steps and choices made by the authors? Furthermore, could you please provide a rationale behind the selection of the procedures and scoring system? It would be helpful to understand the reasoning behind the choices made.
9. The employed statistical methods for group comparison, namely t-test and chi-squared test, are mentioned in the study. However, it is imperative to specify the groups that were compared using these methods. This information should be included in the methodology section to provide clarity and transparency in the study design.
10. In table 2 please clarify where you get Academic performance scores. It should be mentioned in the methods section.
11. on page 12, line 287, authors mentioned that the present study was the first to explore the issue among medical students. However, there was no elaboration provided on the unique academic circumstances of these students that could potentially influence the results, both in the introduction and discussion sections. It would be beneficial to consider including a discussion on this topic to provide more clarity on the study's findings.
12. Please note that in page 12, line 294, the statement "High levels of neuroticism, low levels of extraversion, and ineffective coping strategies characteristic of Type D" refers to the big five personality traits, rather than Type D. It would be appropriate to discuss negative affectivity (NA) and social inhibition (SI) instead.
13. On page 13, line 335 of the discussion section, it is suggested that interventions to reduce negative affect and social inhibition in nurses with Type D personality should be developed and implemented. Additionally, it is recommended that such interventions be encouraged for female nursing students. The reasoning behind this specific recommendation is unclear.
14. The discussion section contains a considerable amount of extraneous information while the primary findings are inadequately explained. For instance, there is a lack of explication of coping strategies in both the introduction and discussion sections.
Comments on the Quality of English LanguageThe manuscript should be checked to be grammatically error-free. E.g., Abstract line 15 “IIn students, the presence of cognitive dysfunction may additionally adversely affect academic and psycho-emotional problems.”
Author Response
This study aimed to examine the influence of type D personality and coping strategies on cognitive functioning in medical students. The current study is on a topic of relevance and general interest to the readers of the journal. However, several issues need to be addressed:
Answer:
We are grateful to the reviewer for his careful review of our manuscript and helpful comments that pointed us to ways to improve the text of our manuscript.
- The background section provides adequate information. However, there is an absence of the study's significance or research gap. To address this issue, it is suggested to reduce the length of the background and allocate more space to the existing research gap. This approach will help to emphasize the importance and relevance of the study, thereby making it more compelling and informative.
Answer:
Thank you for the suggestion, we have significantly adjusted the Introduction section and focused on the problems of students.
- The manuscript title mentions Coping Strategies as an independent variable to explore its impact on cognitive functioning. However, the introduction, results, and discussion presented in the manuscript solely pertain to type D, leaving no indication of any findings related to Coping Strategies.
Answer:
We are grateful to the reviewer for his comment. We have added information about coping strategies in the Introduction, Results, and Discussion sections.
- The introductory section appears to be inadequately written, and the subject matter appears to lack coherence and logic. The lack of coherence may have a negative impact on the overall quality of the work, and could potentially detract from the intended message. It is recommended that revisions be made to the introductory section to enhance its clarity, cohesiveness, and logical flow. By doing so, the overall quality and impact of the work may be improved. The initial statement appears to be merely informative and lacks a clear backdrop for a specific hypothesis.
Answer:
Thank you for the suggestion, we have significantly adjusted the Introduction section and focused on the problems of students.
- The authors seem to have put an excessive focus on cardiovascular disease in their research, which may detract from their primary subject of medical students. It would be beneficial for them to include relevant studies that focus on the unique condition of healthy individuals, particularly students, to provide a thorough background for their research. Without a clear research hypothesis and questions, readers may struggle to understand the purpose of the study. The manuscript could benefit from a well-defined hypothesis and participants that align with the focus on medical students, rather than CVD patients.
Answer:
We are grateful to the reviewer for his comment. We finalized the Introduction section, added information about students' problems and formulated research hypotheses.
- In the methods section, it is recommended to include information regarding the validity and reliability of the measurement instruments used.
Answer:
We have added information about Cronbach’s alpha for the Russian versions of DS14 and questionnaire “The Coping Strategy Indication” to the text of the manuscript.
- Please elaborate on the concept of individual meta-analysis as mentioned in page 2, line 47.
Answer:
Thanks to the reviewer for this question. Because the reviewers recommended that information on type D in cardiovascular disease be reduced, we have removed this reference from the text of the manuscript. Therefore, there is probably no point in commenting in more detail on the concept of this meta-analysis.
- It is important to acknowledge that a significant difference in participation between genders has been observed. This finding must be justified and properly acknowledged as a potential limitation of the study.
Answer:
Thanks to the reviewer for the comment. We have added the following text to the Study Limitations section:
It is important to acknowledge that a significant difference in participation between genders has been observed. This is quite understandable, there are specific features of the gender composition of medical university students - in general, there are more female students than male students. Also, the greater involvement of girls in scientific projects may be explained by their socio-psychological characteristics. Female medical university students are characterized by such traits as conscientiousness, responsibility, and commitment, which explains their educational motivation and greater participation in scientific projects compared to male students. Nevertheless, the significant difference in participation between genders with a predominance of female students is a potential limitation of the study.
- It is imperative to acknowledge the origin of the procedure being discussed. Can we expect a replication study similar to the references provided in the following paragraph? Alternatively, are the steps and choices made by the authors? Furthermore, could you please provide a rationale behind the selection of the procedures and scoring system? It would be helpful to understand the reasoning behind the choices made.
Answer:
The psychophysiological software and hardware complex “Status PF” was developed on the basis of Kemerovo State University. This complex is successfully used in the diagnosis of cognitive impairment in both healthy individuals and patients with cardiovascular diseases. In particular, using this technique, postoperative cognitive dysfunction is assessed, as well as the effectiveness of treatment and resuscitation measures. The accumulated results of studies conducted in our center indicate the reliability of the information obtained when assessing cognitive functions using this complex.
- The employed statistical methods for group comparison, namely t-test and chi-squared test, are mentioned in the study. However, it is imperative to specify the groups that were compared using these methods. This information should be included in the methodology section to provide clarity and transparency in the study design.
Answer:
We have clarified information about the groups in the Methods section:
"Student's t-test, Mann-Whitney test, and χ2 (chi-square) test were used to compare two groups (with type D and without type D)"
- In table 2 please clarify where you get Academic performance scores. It should be mentioned in the methods section.
Answer:
We have added the following text to the Methods section:
Academic performance was assessed by calculating grade point average. Calculation of the average score was done by adding the exam scores for each subject in the session and dividing by the total number of exams
- on page 12, line 287, authors mentioned that the present study was the first to explore the issue among medical students. However, there was no elaboration provided on the unique academic circumstances of these students that could potentially influence the results, both in the introduction and discussion sections. It would be beneficial to consider including a discussion on this topic to provide more clarity on the study's findings.
Answer:
We have added information about medical student concerns to sections of the manuscript
- Please note that in page 12, line 294, the statement "High levels of neuroticism, low levels of extraversion, and ineffective coping strategies characteristic of Type D" refers to the big five personality traits, rather than Type D. It would be appropriate to discuss negative affectivity (NA) and social inhibition (SI) instead.
Answer:
Thank you, we have made a correction to the text
- On page 13, line 335 of the discussion section, it is suggested that interventions to reduce negative affect and social inhibition in nurses with Type D personality should be developed and implemented. Additionally, it is recommended that such interventions be encouraged for female nursing students. The reasoning behind this specific recommendation is unclear.
Answer:
Thank you for your comment. We have made a significant correction to the Discussion section
- The discussion section contains a considerable amount of extraneous information while the primary findings are inadequately explained. For instance, there is a lack of explication of coping strategies in both the introduction and discussion sections.
Answer:
Thank you for your comment. We have made a significant correction to the Discussion section
Comments on the Quality of English Language
The manuscript should be checked to be grammatically error-free. E.g., Abstract line 15 “IIn students, the presence of cognitive dysfunction may additionally adversely affect academic and psycho-emotional problems.”
Answer:
Thank you, we have made a correction to the text
Reviewer 3 Report
Comments and Suggestions for Authors
The study deals with a very important research field. The abstract contains the necessary information. The usefulness and the potential for prevenction should be better emphasised in the abstract and in the study (Why are we investigating this and what will be the benefits of using the data?).
The theoretical background is adequate, but more emphasis should be placed on the links between the different earlier studies. The description of the sample is terse and should be more detailed – we do not know anything about their background, although this would be useful for the main analysis. There should also be a description of what is being done about the very different numbers of people between the sexes in the analysis. This should also be discussed in more detail in the theoretical section (involvement of men and women in the area under study).
In the case of the measurement instruments, examples of the statements in the questionnaires should be given. It would be important to give the statistical indicators of the questionnaires. The relationship between the research objectives and the questionnaires chosen should be explained in more detail. These could be justified in the theoretical background.
The statistical tests are adequate, but I wonder why they were carried out. What do they tell us about the sample? Is it worth examining the sex difference? Is it worth analysing the relevant variables along the background variables? In my opinion, without knowledge of other psychological characteristics, it is difficult to gather useful data from the correlational studies that have been done, or even to use them for anything.
Overall, I see that appropriate statistical methods have been used, but the phenomenon and its deeper meaning is lost behind the statistics.
Comments on the Quality of English LanguageNone
Author Response
Answer:
We are grateful to the reviewer for his careful review of our manuscript and helpful comments that pointed us to ways to improve the text of our manuscript.
The study deals with a very important research field. The abstract contains the necessary information. The usefulness and the potential for prevenction should be better emphasised in the abstract and in the study (Why are we investigating this and what will be the benefits of using the data?).
Answer:
Thank you for your comment. We have made a significant correction to the Introduction section
The theoretical background is adequate, but more emphasis should be placed on the links between the different earlier studies. The description of the sample is terse and should be more detailed – we do not know anything about their background, although this would be useful for the main analysis. There should also be a description of what is being done about the very different numbers of people between the sexes in the analysis. This should also be discussed in more detail in the theoretical section (involvement of men and women in the area under study).
Answer:
Thank you for your comment. We have made significant corrections to the manuscript and added information to the Introduction, Discussion, and Limitations of the study sections.
In the case of the measurement instruments, examples of the statements in the questionnaires should be given. It would be important to give the statistical indicators of the questionnaires. The relationship between the research objectives and the questionnaires chosen should be explained in more detail. These could be justified in the theoretical background.
Answer:
We have added information about Cronbach’s alpha for the Russian versions of DS14 and questionnaire “The Coping Strategy Indication” to the text of the manuscript. More detailed information about the questionnaires and questionnaires used was presented in our previously published works, so we did not repeat them in this article.
The statistical tests are adequate, but I wonder why they were carried out. What do they tell us about the sample? Is it worth examining the sex difference? Is it worth analysing the relevant variables along the background variables? In my opinion, without knowledge of other psychological characteristics, it is difficult to gather useful data from the correlational studies that have been done, or even to use them for anything.
Answer:
In the revised version of the Discussion section, we examined in more detail the results of our research (coping strategies and type D, the “type D effect,” etc.), and compared them with literature data. We hope that in this form the purpose of using statistical methods will become clearer.
Also we added to Method section the text: “To identify a possible association of the cognitive tests indicators (“Concentration of Attention” test and “Functional mobility of nervous processes” test) with components of personality type D (Type D, ZscoreNA, ZscoreSI, and zNA × zSI) , we additionally performed linear regression analysis’.
Overall, I see that appropriate statistical methods have been used, but the phenomenon and its deeper meaning is lost behind the statistics.
Answer:
In the revised version of the Discussion section, we examined in more detail the results of our research (coping strategies and type D, the “type D effect,” etc.), and compared them with literature data. We hope that in this form the purpose of using statistical methods will become clearer.
Round 2
Reviewer 1 Report
Comments and Suggestions for Authors
Dear Authors,
I am pleased to know that you have taken into account the recommendations made in my first review of this manuscript.
I consider that the most significant changes have been made and the study is ready for publication.
Congratulations on your research,
Best regards
Reviewer 2 Report
Comments and Suggestions for Authors Thank you for the opportunity to review the revised manuscript titled “The influence of personality type D and coping strategies on cognitive functioning in students”. I appreciate the time and effort that the authors have dedicated to addressing the comments and suggestions from the previous round of review. The authors have made significant revisions to the manuscript, which have improved its clarity, quality, and contribution. The revised manuscript is well-written, well-structured, and well-supported by the evidence. I recommend the acceptance of the manuscript, and I congratulate the authors on their excellent work and look forward to seeing their paper published in this journal. Sincerely,